# SDP4Bit: Toward 4-bit Communication Quantization in Sharded Data Parallelism for LLM Training

**Jinda Jia**[*]
Indiana University
jindjia@iu.edu

**Cong Xie**[*]
ByteDance Inc.
cong.xie@bytedance.com

**Hanlin Lu**
ByteDance Inc.
hanlin.lu@bytedance.com

**Daoce Wang**
Indiana University
daocwang@iu.edu

**Hao Feng**
Indiana University
haofeng@iu.edu

**Chengming Zhang**
University of Houston
czhang59@cougarnet.uh.edu

**Baixi Sun**
Indiana University
sunbaix@iu.edu

**Haibin Lin**
ByteDance Inc.
haibin.lin@bytedance.com

**Zhi Zhang**
ByteDance Inc.
zhangzhi.joshua@bytedance.com

**Xin Liu**
ByteDance Inc.
liuxin.ai@bytedance.com

**Dingwen Tao**
Indiana University
ditao@iu.edu

## Abstract

Recent years have witnessed a clear trend towards language models with an ever-increasing number of parameters, as well as the growing training overhead and memory usage. Distributed training, particularly through Sharded Data Parallelism (ShardedDP) which partitions optimizer states among workers, has emerged as a crucial technique to mitigate training time and memory usage. Yet, a major challenge in the scalability of ShardedDP is the intensive communication of weights and gradients. While compression techniques can alleviate this issue, they often result in worse accuracy. Driven by this limitation, we propose **SDP4Bit** (Toward 4Bit Communication Quantization in Sharded Data Parallelism for LLM Training), which effectively reduces the communication of weights and gradients to nearly 4 bits via two novel techniques: quantization on weight differences, and two-level gradient smooth quantization. Furthermore, SDP4Bit presents an algorithm-system co-design with runtime optimization to minimize the computation overhead of compression. In addition to the theoretical guarantees of convergence, we empirically evaluate the accuracy of SDP4Bit on the pre-training of GPT models with up to 6.7 billion parameters, and the results demonstrate a negligible impact on training loss. Furthermore, speed experiments show that SDP4Bit achieves up to $4.08\times$ speedup in end-to-end throughput on a scale of 128 GPUs.

## 1 Introduction

Large Language Models (LLMs) are increasingly utilized across various applications, leading to a trend toward larger model sizes. This expansion in model size significantly escalates training overheads, making the process more costly and resource-intensive. To mitigate the time-consuming

---

[*]Equal Contribution.

38th Conference on Neural Information Processing Systems (NeurIPS 2024).

nature of training LLMs, it is common to employ multiple GPUs in a data-parallel configuration. However, naive Data Parallelism (DP) necessitates that each GPU replicates the entire optimizer states, a strategy often impractical due to the limited memory capacity of individual GPUs. This limitation becomes particularly critical with the substantial size of modern LLMs.

Sharded Data Parallelism (ShardedDP) evolves from naive DP to reduce the memory footprint by sharding optimizer states among GPUs. However, the sharding mechanism significantly changes the communication pattern of DP, which brings up new challenges in system optimization. As a result, ShardedDP suffers from heavy communication overheads of both weights and gradients, particularly when inter-node bandwidth is limited. This can significantly increase the end-to-end (E2E) training time, especially when using a small gradient accumulation step.

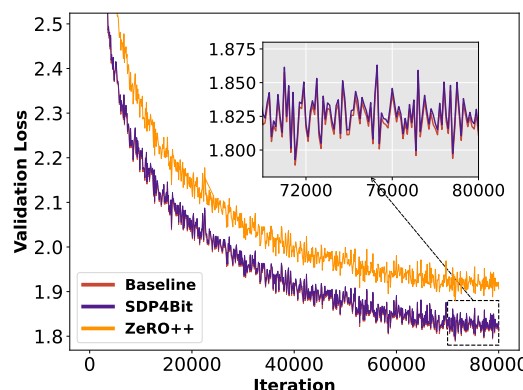

Figure 1: Training validation loss for GPT-6.7B; SDP4Bit is closely aligned with full precision training.

Quantization is a widely used strategy to reduce the communication overhead of naive DP, albeit with some accuracy loss. Unfortunately, few prior studies have specifically addressed the issue of communication reduction in ShardedDP. Recently, QSDP [18] and ZeRO++ [32] attempted to quantize the communication of ShardedDP to Int4. However, when pushing the communication ratio to its limits, both QSDP and ZeRO++ fail to maintain comparable training loss to the baseline. Furthermore, ZeRO++ lacks theoretical convergence guarantees, and QSDP is limited to one specific quantizer called "random shift" and strong assumptions. Thus, *there is no effective solution to reduce ShardedDP's communication to nearly 4 bits without compromising the training loss*.

To address these issues, this paper proposes a novel communication reduction strategy, **SDP4Bit**. SDP4Bit comprises two main techniques: (1) **Quantization on Weight Differences**: Instead of directly quantizing weights, we apply 4-bit quantization to compress the weight differences between current and previous iterations; (2) **Two-Level Gradient Smooth Quantization**: We apply 8-bit quantization to intra-node gradients and 4-bit quantization to inter-node gradients, with Hadamard Transform for smoothing the outliers. To the best of our knowledge, SDP4Bit is the first work to successfully reduce both gradients and weights to nearly 4 bits without compromising training accuracy. As shown in Figure 1, the training validation loss for GPT-6.7B using SDP4Bit is closely aligned with full precision training. Our main contributions are summarized as follows:

- We propose a low-bit (i.e., nearly 4-bit) communication reduction strategy for ShardedDP that preserves E2E training accuracy.
- We establish a convergence guarantee for the proposed strategy, showing the same convergence rate as the ordinary Stochastic Gradient Descent (SGD), with extended choices of biased compressors and weaker assumptions compared to the previous theoretical results.
- We implement our method within the Megatron-LM framework and enhance it with runtime optimizations such as buffer reuse, operation pruning, and kernel fusion.
- Our results validate that SDP4Bit successfully compresses the communication of weights and gradients to nearly 4 bits, with a negligible impact on final loss. Notably, compared to non-quantized baseline, it achieves 4.08× speedup for a GPT-18B model trained on 128 H800 GPUs.

## 2 Preliminaries

### 2.1 Sharded Data Parallelism

Sharded Data Parallelism (ShardedDP) modifies traditional Data Parallelism (DP) to reduce the memory footprint per GPU. Unlike traditional DP, which duplicates high-precision optimizer states (typically including model weights and momentum variables in Float32) on each GPU, ShardedDP partitions them across all GPUs. Each GPU manages $\frac{1}{P}$ of the optimizer states, hence reducing the corresponding memory footprint by a factor of $\frac{1}{P}$, where $P$ represents the number of GPUs involved.

| **Algorithm 1** QSDP / ZeRO++ | **Algorithm 2** Megatron-LM with SDP4Bit |
|---|---|
| **Require:** worker: $p$, weight in shard $p$: $w[p]$, local gradient on worker $p$: $g_{model}^p$, global gradient in shard $p$: $g_{main}[p]$ | **Require:** worker: $p$, weight in shard $p$: $w[p]$, local gradient on worker $p$: $g_{model}^p$, global gradient in shard $p$: $g_{main}[p]$, weight difference: $d$ |

Algorithm 1:

1: **function** CompressedForwardPass
2:     $\tilde{w}_{main}[p] \leftarrow$ QuantizeWeights($w_{main}[p]$)
3:     $w_{model} \leftarrow$ AllGather($\tilde{w}_{main}[p]$)
4:     $output^p \leftarrow$ ForwardPass($w_{model}, input^p$)
5:     free($w_{model}$)
6: **function** CompressedBackwardPass
7:     $\tilde{w}_{main}[p] \leftarrow$ QuantizeWeights($w_{main}[p]$)
8:     $w_{model} \leftarrow$ AllGather($\tilde{w}_{main}[p]$)
9:     $g_{model}^p \leftarrow$ Gradient($w_{model}, output^p$)
10:     free($w_{model}$)
11:     $\tilde{g}_{model}^p \leftarrow$ QuantizeGradients($g_{model}^p$)
12:     $g_{main}[p] \leftarrow$ ReduceScatter/TwoAlltoAll($\tilde{g}_{model}^p$)
13:     $w_{main}[p] \leftarrow$ Optimizer($g_{main}[p], w_{main}[p]$)

Algorithm 2:

1: **function** CompressedForwardPass
2:     $d[p] = w_{main}[p] - w_{model}[p]$
3:     $\tilde{d}[p] \leftarrow$ QuantizeWeightsDiff($d[p]$)
4:     $d \leftarrow$ AllGather($\tilde{d}[p]$)
5:     $w_{model} \leftarrow w_{model} + d$
6:     $output^p \leftarrow$ ForwardPass($w_{model}, input^p$)
7: **function** CompressedBackwardPass
8:     $g_{model}^p \leftarrow$ Gradient($w_{model}, output^p$)
9:     $g_{main}[p] \leftarrow$ ~~ReduceScatter~~ TLq-HS($g_{model}^p$)
10:     $w_{main}[p] \leftarrow$ Optimizer($g_{main}[p], w_{main}[p]$)

With high-precision model weights sharded across GPUs (referred to as **"main weights"**), an all-gather operation is required to collect the weights for the forward-backward steps (typically in relatively low precision, such as Float16, referred to as **"model weights"**). For gradient synchronization, a reduce-scatter operation is performed before the optimization steps to ensure that each GPU has the corresponding shard of averaged gradients. In summary, each iteration necessitates an all-gather for weights and a reduce-scatter for gradients.

Driven by the need to train larger models within the constraints of GPU memory, ShardedDP is incorporated into several popular training frameworks, including Megatron-LM (Distributed Optimizer), DeepSpeed (ZeRO), and PyTorch (FSDP), each with slightly different implementation strategies. Notably, ZeRO-3 and FSDP release the collected weights after each computation to enhance memory efficiency, necessitating additional weight collective communication during the backward pass. Conversely, ZeRO-2 and Megatron-LM retain the collected weights throughout, thus eliminating the need for weight collection during the backward pass. Our weight reduction strategy is particularly well-suited for Megatron-LM, as it maintains a full model's weights at all times (see Algorithm 2). Additionally, Megatron-LM provides flexible parallelism support, such as tensor parallelism, which partitions models vertically to alleviate memory limitations. This approach enables the training of larger models compared to DeepSpeed.

## 2.2 Quantization

Quantization is a commonly used strategy in data compression. In this paper, we explore symmetric linear (integer) quantization due to its low overhead and latency. It is defined as follows:

$$x_{\text{int}} = \text{round}\left(\frac{x}{s} \cdot (2^{k-1} - 1)\right), \quad s = \max(x),$$

where $k$ represents the bit-width of the quantized values, and $s$ is referred to as "scales".

Additionally, group-wise quantization [25] is employed to minimize quantization error by dividing the data into multiple groups and quantizing each group individually. This approach results in a lower compression ratio due to the need to store additional scales.

## 2.3 Collective Reduction Communication with Quantization

State-of-the-art (SOTA) collective communication libraries (e.g., NCCL, Gloo) employ a ring-based algorithm for its optimal bandwidth [21]. This algorithm executes reduce-scatter operations across $P - 1$ rounds, during which each GPU sends local data and aggregates the received data. When quantization is applied, this necessitates $P - 1$ rounds of quantization and dequantization, potentially leading to error propagation and increased latency [11]. Some strategies replace reduce-scatter with all-to-all communication, but this increases inter-node communication, typically with lower bandwidth.

ZeRO++ [32] modifies this approach by substituting the conventional reduce-scatter (used by QSDP) with two all-to-all operations (shown in Algorithm 1, with different colors to distinguish ZeRO++

from QSDP). The first operation is confined within each node, and post-reduction, the data size is diminished to $\frac{1}{N}$, where $N$ is the number of GPUs per node. The subsequent all-to-all operation occurs between GPUs across different nodes that share the same local rank. In ZeRO++, each all-to-all operation follows a 4-bit quantization step to minimize communication data size.

As shown in Figure 5, while this method efficiently integrates quantization into reduce-scatter without augmenting inter-node communication, *the repeated 4-bit quantization steps can accumulate quantization errors, potentially leading to suboptimal training outcomes*.

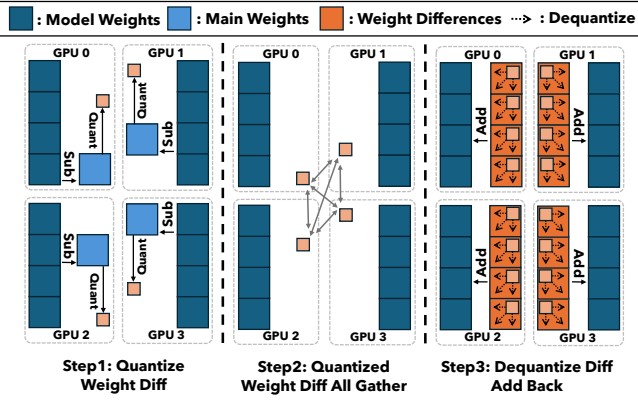

Figure 2: Communication of quantized weight differences.

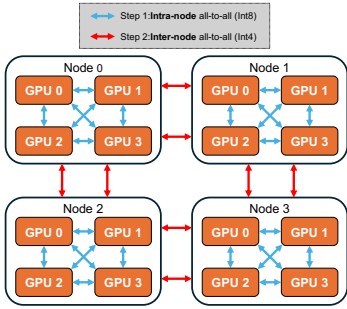

Figure 3: Two-level gradient quantization: 8-bit intra-node and 4-bit inter-node quantization.

## 3 Methodology

### 3.1 Quantization on Weight Differences *(qWD)*

As discussed in Section 2.1, ShardedDP requires each GPU to send/receive updated weights (main weights) to/from other GPUs in each iteration. *However, weights generally exhibit a wide range and directly applying 4-bit quantization leads to significant quantization errors*. Even with groupwise quantization, a gap in E2E training loss compared to full precision remains despite using small group sizes.

**qWD:** To address this issue, we quantize weight differences instead of the original weights during communication. As illustrated in Figure 2 and Algorithm 2, after the optimizer step, each GPU calculates the

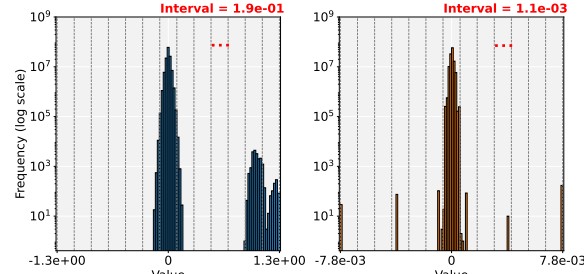

Figure 4: Histogram of (a) weights and (b) weight differences. Each vertical dashed line represents a quantization level corresponding to a 4-bit quantization lattice.

differences between the main weights and the model weights. These differences are then quantized and all-gathered across all GPUs. After all-gathering, each GPU dequantizes the received data to obtain the weight differences. These differences are then added to the model weights to obtain the updated weights.

There are two main benefits to applying quantization to weight differences.

**1)** In practice, weight differences are generally easier to quantize. As shown in Figure 4, weight differences are more uniformly distributed in a smaller range compared to the weights themselves, resulting in smaller errors for INT4 quantization. Furthermore, since intuitively the magnitudes of weight differences are smaller than those of weights themselves (informally supposed that $\|\delta w_t\| = \|w_t - w_{t-1}\| < \|w_t\|$) and the relative quantization errors are similar between weights and weight differences (informally supposed that $\frac{\|q(\delta w_t) - \delta w_t\|}{\|\delta w_t\|} \approx \frac{\|q(w_t) - w_t\|}{\|w_t\|}$), the

weight differences compression potentially has a smaller error relative to the weights themselves: $\frac{\|q(\delta w_t) - \delta w_t\|}{\|w_t\|} \lesssim \frac{\|q(w_t) - w_t\|}{\|w_t\|}$, where $q(\cdot)$ is the quantization function.

**2)** In theory, weight differences compression improves convergence compared to naive weight compression. When extended to biased compressors, we present theoretical guarantees for convergence at the same rate as ordinary SGD, as detailed in Section 4.2. In contrast, we demonstrate that using biased compressors directly on weights can lead to convergence failure, as illustrated in an example in Section 4.1. This proves that biased compressors are not compatible with QSDP or ZeRO++.

## 3.2 Two-Level Gradient Smooth Quantization

### 3.2.1 Two-level Gradient Quantization *(TLq)*

As discussed in Section 2.3, the 2-step all-to-all communication strategy benefits gradient communication when quantization is applied. However, it also introduces error accumulation due to consecutive 4-bit quantization steps, necessitating additional rounds of training and communication that diminish the per-iteration communication savings. We observe that *applying ULq, which quantizes gradeints to extremely low precision, such as 4-bit, leads to noticeable deviations in training loss* compared to full-precision training, as illustrated in Figure 5.

*TLq:* Instead of employing a global 4-bit quantization strategy for both inter-node and intra-node all-to-all communications, we propose a two-level precision quantization strategy. This approach balances performance and accuracy by enhancing accuracy without introducing additional overhead. For intra-node all-to-all communication, gradients are quantized to INT8 before sending. After receiving the data, each GPU dequantizes the received data back to the full precision (i.e., FP32) for local reduction. This reduced data is then quantized to INT4 to minimize inter-node communication overhead. The detailed methodology is depicted in Figure 3. Since the two all-to-all operations utilize different network bandwidths, their communications can be effectively overlapped (see Table 4).

### 3.2.2 *TLq* with Hadamard Smoother *(TLq-HS)*

While *TLq* brings the training loss closer to the baseline, it does not achieve perfect alignment. In gradient quantization, *outliers can significantly amplify quantization errors*. Although group-wise quantization isolates outliers to minimize their impact on the precision of values in other groups, the values within the same group remain affected.

*TLq-HS:* To mitigate the outlier issue, we apply Hadamard transform [9] to the gradients before quantization. The Hadamard transform, a specific type of generalized Fourier transform, exhibits properties such as $H = H^T$ and $H \cdot H^T = I$, distributing outlier information across nearby elements and effectively smoothing them out. For a detailed description of the methodology, see Algorithm 3.

## 3.3 Performance Optimizations in Implementation

**Optimizing Memory Efficiency for Weight Differences Computation:** After the all-gather communication, each GPU receives weight differences from the others, which are then added to the model weights to update them. As discussed in Section 2.1, with ShardedDP enabled in Megatron-LM, each GPU maintains a complete copy of the model weights for forward and backward computation. This contrasts with ZeRO, where model weights are released after computation. By reusing these locally stored model weights in Megatron-LM, our implementation eliminates the need for additional buffers to retain model weights for calculating the differences, thus enhancing memory efficiency.

---

**Algorithm 3** TLq with Hadamard Smoother

---

**Require:** gradient $grad$
1: **function** TLq-HS
2:     $\hat{g} \leftarrow$ Hadamard($grad$)
3:     $\hat{q}g_{8bit} \leftarrow$ Quantize8Bit($\hat{g}$)
4:     $list(\hat{q}g_{8bit\_intra}) \leftarrow$ **IntraAlltoAll**($\hat{q}g_{8bit}$)
5:     $list(\hat{g}_{intra}) \leftarrow$ Dequantize($list(\hat{q}g_{8bit\_intra})$)
6:     ~~$list(\hat{g}_{intra}) \leftarrow$ Hadamard($list(\hat{g}_{intra})$)~~
7:     $\hat{g}_{reduced} \leftarrow$ Reduction($list(\hat{g}_{intra})$)
8:     ~~$\hat{g}_{reduced} \leftarrow$ Hadamard($\hat{g}_{reduced}$)~~
9:     $\hat{q}g_{4bit} \leftarrow$ Quantize4Bit($\hat{g}_{reduced}$)
10:    $list(\hat{q}g_{4bit\_inter}) \leftarrow$ **InterAlltoAll**($\hat{q}g_{4bit}$)
11:    $list(\hat{g}_{inter}) \leftarrow$ Dequantize($list(\hat{q}g_{4bit\_inter})$)
12:    $\hat{g}_{final\_reduced} \leftarrow$ Reduction($list(\hat{g}_{inter})$)
13:    $g_{final\_reduced} \leftarrow$ Hadamard($\hat{g}_{final\_reduced}$)

---

**Simplifying Hadamard Transforms:** In a naive implementation, the Hadamard transform would be applied at each step before quantization and after dequantization. However, by leveraging the orthogonality of the Hadamard transform, i.e. $H \cdot H = I$, we omit the transform after the intra-node all-to-all dequantization (Algorithm 3, Line 6) and before the inter-node quantization (Algorithm 3, Line 8). Furthermore, by utilizing the distributive property, i.e., $\sum_i H g_i = H \sum_i g_i$, we move the second Hadamard transform from after the inter-node dequantization (Algorithm 3, Line 11) to after the final reduction (Algorithm 3, Line 12). These simplifications reduce the unnecessary computational overhead associated with repeated Hadamard transforms.

**Fusing GPU Kernels with Group Size Alignment:** To mitigate additional data movement from global memory—which typically exhibits the slowest memory bandwidth—, we fuse the Hadamard transform with the (de)quantization operations into a single CUDA kernel. This fusion allows the operations to run nearly as fast as the quantization operation alone. It is worth noting that, for this fusion to be efficient, there must be an alignment between the two. Specifically, the size of the quantization group must be divisible by the size of the Hadamard matrix, ensuring that memory traffic remains within the kernel block. We choose $H$ to be small (e.g., 32×32) because, at this size, the transform operation on the GPU is typically memory-bound and incurs minimal overhead. While larger $H$ sizes offer better smoothing capabilities, we find that a 32×32 matrix is sufficient to effectively smooth outliers in gradients.

# 4 Theoretical Analysis

## 4.1 Counterexample of Biased Weight Compression

One of the advantages of our proposed weight difference compression is the compatibility to both biased and unbiased compressors. Note that using biased compressors directly on weight compression incurs issues in convergence under standard assumptions, as avoided in QSDP [18] or ZeRO++ [32]. We illustrate such issues in the following toy example.

**Counterexample 4.1.** Consider a least square problem with $w^* = (0,0)^\top$: $\min_{w \in \mathbb{R}^2} \left[ f(w) = \|w\|^2 \right]$, and stochastic gradient $g(w) = (4w_1, 0)^\top$ with probability 0.5 and $g(w) = (0, 4w_2)^\top$ with probability 0.5, thus $\mathbb{E}[g(w)] = \nabla_w f(w)$. We use the initial value $w_{init} = (1, -1)^\top$, the learning rate $\eta < 0.125$, and the following nearest ternary quantizer: $s = \max(|w|), q(w) = round(w/s) * s$, where $|\cdot|$ is element-wise absolute value, and $round(\cdot)$ quantizes each element to the nearest value in $\{-1, 0, 1\}$. It is easy to check that for SGD under such settings, the weights before quantization will be either $(1 - 4\eta, -1)^\top$ or $(1, -1 + 4\eta)^\top$, resulting in $q(w) = (1, -1)^\top$, which means that SGD with ternary weight quantization gets stuck at the initial value in this case, while SGD without weight quantization and SGD with weight difference quantization both converge to the optimal.

## 4.2 Convergence Analysis

To theoretically analyze the convergence of our distributed training algorithm with communication compression, we focus on the following SGD variant with gradient compression and weight difference compression. We use SGD to solve the following optimization problem: $f^* = \min_w f(w)$, where $f(w)$ is the objective function, $w \in \mathbb{R}^d$ is the model parameter.

---

**Algorithm 4** SGD with SDP4Bit

1: Initialize main parameter weights $w_0$
2: Initialize compressed parameter weights $\tilde{w}_0 \leftarrow w_0$
3: **for all** iteration $t \in [T]$ **do**
4:  Compute gradient: $g_{t-1} = \nabla f(\tilde{w}_{t-1}; \zeta_{t-1})$
5:  Compress gradient: $\tilde{g}_{t-1} = \mathcal{U}_g(g_{t-1})$
6:  Update main weights: $w_t \leftarrow w_{t-1} - \eta \tilde{g}_{t-1}$
7:  Compress weight difference: $\tilde{\Delta}_t = \mathcal{C}_w(w_t - \tilde{w}_{t-1})$
8:  Update compressed weights: $\tilde{w}_t \leftarrow \tilde{w}_{t-1} + \tilde{\Delta}_t$
9: **end for**

---

Note that we use unbiased compressors for gradient reduction, and arbitrary (potentially biased) compressors for weight collection. We formally define these two classes of compressors as follows.

**Definition 4.1** (Unbiased $\kappa$-approximate compressor [1]). An operator $\mathcal{U} : \mathbb{R}^d \to \mathbb{R}^d$ is a $\kappa$-approximate compressor for $\kappa \geq 0$ if $\mathbb{E}[\mathcal{U}(v)] = v$ and $\mathbb{E}\|\mathcal{U}(v) - v\|^2 \leq \kappa\|v\|^2, \forall v \in \mathbb{R}^d$.

**Definition 4.2** ($\delta$-approximate compressor [13]). An operator $\mathcal{C} : \mathbb{R}^d \to \mathbb{R}^d$ is a $\delta$-approximate compressor for $\delta \in [0, 1]$ if $\mathbb{E}\|\mathcal{C}(v) - v\|^2 \leq (1 - \delta)\|v\|^2, \forall v \in \mathbb{R}^d$.

*Remark* 4.1. Note that, in a certain sense, the class of $\delta$-approximate compressors contains the class of unbiased compressors. It is easy to check that any $\kappa$-approximate unbiased compressor $\mathcal{U}$ can be converted to a $\frac{1}{1+\kappa}$-approximate biased compressor $\mathcal{C}(v) = \frac{1}{1+\kappa}\mathcal{U}(v)$. Furthermore, the class of $\delta$-approximate compressors typically provides more options such as top-$k$ sparsifiers, and top-$k$ low-rank compressors. Thus, we consider arbitrary (biased or unbiased) $\delta$-approximate compressors for weight compression in our theoretical analysis.

*Remark* 4.2. For distributed training with $P$ workers, we define the compressed gradient as $\tilde{g}_t = \mathcal{U}_g(g_t) = \frac{1}{P}\sum_{i\in[P]}\mathcal{U}'_g(g_{t,i})$, where $g_t = \frac{1}{P}\sum_{i\in[P]}g_{t,i}$, and $g_{t,i}$ is the stochastic gradient from the $i$th worker in $t$ iteration. We assume that $\mathcal{U}_g$ is an unbiased $\kappa$-approximate compressor of the average gradient $g_t$.

**Assumption 4.1.** (Smoothness) We assume that $f(x)$ is $L$-smooth: $\|\nabla f(x) - \nabla f(y)\| \leq L\|x - y\|, \forall x, y \in \mathbb{R}^d$, which implies $f(y) - f(x) \leq \langle\nabla f(x), y - x\rangle + \frac{L}{2}\|y - x\|^2$.

**Assumption 4.2.** For any stochastic gradient $\nabla f(w; \zeta)$, where $\zeta$ is an independent random sample, we assume unbiasedness $\mathbb{E}[\nabla f(w; \zeta)|w] = \nabla f(w)$, and bounded variance $\mathbb{E}[\nabla f(w; \zeta) - \nabla f(w)\|^2|w] \leq \rho\|\nabla f(w)\|^2 + \sigma^2$ ([27], Assumption 3).

We derive the following error bounds on the convergence of SDP4Bit under the above assumptions. All proofs can be found in Appendix A.

**Theorem 4.1** (Convergence error bound). *For arbitrary non-convex function under Assumption 4.1 and Assumption 4.2, taking learning rate $\eta \leq \frac{1}{10L\left(\frac{2}{\delta}+\rho\kappa+\rho+\kappa\right)}$, Algorithm 4 converges to a critical point with the following error bound:*

$$\frac{\sum_{t=0}^{T}\mathbb{E}[\|\nabla f(\tilde{w}_t)\|^2]}{T+1} \leq \frac{80L\left(\frac{2}{\delta}+\rho\kappa+\rho+\kappa\right)(f(w_0)-f^*)}{T+1} + 4\sigma\sqrt{\frac{(11-\delta)(\kappa+1)L(f(w_0)-f^*)}{T+1}}.$$

*Remark* 4.3. Note that compared to QSDP [18], our convergence analysis does not require Polyak-Łojasiewicz condition or the specific choice of weight quantization (random shift). In other words, Theorem 4.1 shows that our proposed algorithm has the same $\mathcal{O}\left(\frac{1}{\sqrt{T}}\right)$ convergence rate as ordinary SGD for general non-convex functions, but under much weaker assumptions compared to QSDP.

# 5 Evaluation

## 5.1 Experimental Setup

**Hardware:** The experiments are conducted on two different clusters to evaluate SDP4Bit across varying network environments: **1)** 16 nodes, each node equipped with 4 Nvidia A100-SXM4-40GB GPUs. All nodes are interconnected with a 100 Gbps Slingshot10 network, providing slower inter-node bandwidth. **2)** 16 nodes, each node equipped with 8 Nvidia H800-SXM5-80GB GPUs. Each node is connected using 8 InfiniBand links, achieving a total bandwidth of 3.2 Tbps, providing higher inter-node bandwidth.

**Baselines:** We use BFloat16/Float32 (weights/gradients) mixed-precision in Megatron-LM [26] as our basic *Baseline* for both accuracy and E2E throughput analysis. Within each set of experiments, we ensure consistent hyper-parameters to ensure fairness. Detailed parameters are provided in Appendix D. Additionally, we implement another baseline for comparison in Megatron-LM, using the same quantization strategy in ZeRO++, employing 4-bit quantization for both weights (group-wise weight quantization, refered to as *qW*) and gradients (twice all-to-all with uniform level 4-bit quantization, refer to as *ULq*).

**Dataset and Models:** To demonstrate that SDP4Bit does not adversely affect end-to-end training loss, we conduct pre-training on GPT-series [23] models ranging from 125M to 6.7B parameters

Table 1: Final validation loss↓ of pre-training with different quantization strategies.

| GPT Model | Baseline | Weight | | Gradient | | SDP4Bit |
| | | qW | qWD | TLq | TLq-HS | |
| --- | --- | --- | --- | --- | --- | --- |
| 125M | 2.29392 | 2.57312 | 2.29274 | 2.30479 | 2.29528 | 2.29590 |
| 350M | 2.08719 | 2.27405 | 2.08730 | 2.09551 | 2.08912 | 2.08964 |
| 1.3B | 1.92774 | 2.04608 | 1.92881 | 1.95075 | 1.93134 | 1.93238 |

on the Pile dataset [8], using validation loss as the accuracy measure. Each test runs for 80,000 iterations, processing over 40 billion tokens. For throughput evaluation, we select models ranging from 1.3B to 18B parameters, with end-to-end training throughput as the metric. In these tests, the accumulation step is set to 1. Note that model parallel is required for models larger than 6.7B, and different tensor parallel sizes are used on A100 and H800 clusters for models larger than 13B. Please refer to Appendix B Table 8 for detailed model parallel configuration.

## 5.2 Accuracy Evaluation

First, we analyze the impact of SDP4Bit on the accuracy of E2E training. As shown in Table 1, the training results for three different model sizes indicate that the final loss of SDP4Bit is comparable to the baseline, with a maximum increase of only 0.24%. Additionally, Figure 1 details the training curve of GPT-6.7B, demonstrating that the training curve of SDP4Bit perfectly aligns with the baseline. This indicates that *the impact of SDP4Bit on accuracy is negligible*. In contrast, the 4-bit quantization strategy in ZeRO++ (*which directly applies quantization to weights and uniformly uses 4-bit quantization with all-to-all for gradients*) results in significant accuracy degradation.

Next, we break down and analyze each strategy within SDP4Bit. **1) For weight communication reduction,** as shown in Table 1, directly quantizing weights (*qW*) to 4 bits results in a validation loss that increase of up to 12% compared to the baseline. In contrast, our weight difference quantization (*qWD*) method achieves a validation loss nearly identical to the baseline. Notably, we use a consistent quantization group size of 2048 for both tests. **2) For gradient communication reduction,** as shown in Figure 5, applying **U**niform **L**evel **q**uantization *(ULq)*, similar to the method used in ZeRO++, results in a significant gap in the loss compared to the baseline. In comparison, our **T**wo **L**evel **q**uantization *(TLq)* siginificantly mitigate the loss gap between with baseline. Additionally, Figure 6 illustrates the effectiveness of the Hadamard transformation in smoothing outliers. Table 1 and Figure 5 further demonstrate the contribution of the Hadamard smoother to accuracy. Notably, compared to *TLq*, *TLq-HS* further narrows the validation loss gap, making it almost identical to the baseline.

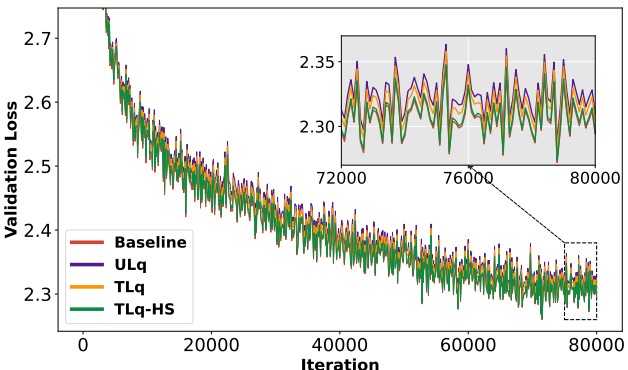 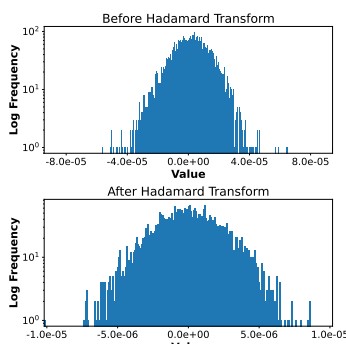

Figure 5: Validation loss comparison for the Baseline, ULq, TLq, and TLq-HS on the GPT-125M model. Uniformly applying 4-bit gradient quantization twice results in a noticeable gap compared to the baseline. In contrast, two-level quantization (8-bit for intra-node and 4-bit for inter-node) mitigates this gap. The Hadamard smoother further reduces the gap, making the loss nearly identical to the baseline.

Figure 6: Comparison of gradient histograms before and after the Hadamard transformation. The transformation reduces the impact of outliers, resulting in a smoother gradient distribution.

Table 2: E2E throughput↑ on different model sizes with std.

| Model Size | 4xA100, 16 nodes (Slingshot 10) | | | 8xH800, 16 nodes (InfiniBand) | | |
|---|---|---|---|---|---|---|
| | Baseline TFLOPs | SDP4Bit TFLOPs | Speedup | Baseline TFLOPs | SDP4Bit TFLOPs | Speedup |
| 1.3B | 24.1 ±0.03 | 57.6 ±0.03 | 2.39× | 69.1 ±0.96 | 106.0 ±2.66 | 1.53× |
| 2.7B | 24.0 ±0.00 | 58.4 ±0.07 | 2.43× | 71.9 ±0.56 | 116.9 ±0.98 | 1.63× |
| 6.7B | 10.8 ±0.00 | 37.1 ±0.00 | 3.44× | 26.2 ±0.33 | 77.9 ±2.43 | 2.97× |
| 13B | 9.7 ±0.04 | 26.0 ±0.03 | 2.68× | 13.9 ±0.17 | 53.5 ±1.36 | 3.85× |
| 18B | 10.2 ±0.00 | 29.8 ±0.04 | 2.92× | 14.5 ±0.07 | 59.2 ±1.37 | 4.08× |

Table 3: Final validation loss↓ of GPT-125M with different group sizes.

| Baseline Val Loss | | 2.29392 | |
|---|---|---|---|
| TLq-HS | Group Size | 64 | 128 | 512 |
| | Val Loss | 2.29537 | 2.29528 | 2.29670 |
| qWD | Group Size | 1024 | 2048 | |
| | Val Loss | 2.29338 | 2.29274 | |
| qW | Group Size | 32 | 128 | 2048 |
| | Val Loss | 2.39580 | 2.44712 | 2.57312 |

Table 4: Performance of Different Quantization Strategies on GPT-1.3B over 32 A100 with standard deviation.

| Quantization Strategy | Grad. Comm. Time (ms) | TFLOPs |
|---|---|---|
| Baseline | 379.3 ±0.34 | 24.4 ±0.00 |
| TLq-HS | 45.9 ±0.03 | 43.3 ±0.05 |
| ULq | 45.0 ±0.03 | 43.3 ±0.04 |
| **SDP4Bit** | 45.8 ±0.03 | 58.5 ±0.07 |
| SDP4Bit (HS w/o fused) | 64.6 ±0.05 | 55.2 ±0.07 |

## 5.3 Throughput Evaluation

Next, we evaluate the improved E2E throughput, measured in FLOPS per second, of SDP4Bit on both hardware platforms. For all tests, the results are averaged over 10 iterations after 20 warm-up iterations. As shown in Table 2, SDP4Bit achieves an E2E training speedup of up to 4.08×. For models with the same model parallel configuration (e.g., 1.3B and 2.7B; 13B and 18B), both the E2E throughput and speedup from SDP4Bit increase as the model size grows due to larger models having higher computational efficiency but also encountering increased communication overhead.

The throughput of the 1.3B, 2.7B, and 6.7B models across the two platforms indicates that SDP4Bit provides a more significant speedup when network bandwidth is lower. This is because lower bandwidth results in higher communication overhead, which SDP4Bit effectively reduces through efficient quantization techniques.

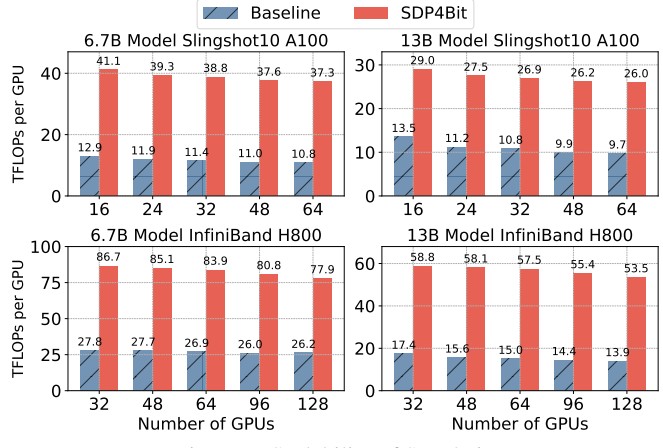

Figure 7: Scalability of SDP4Bit.

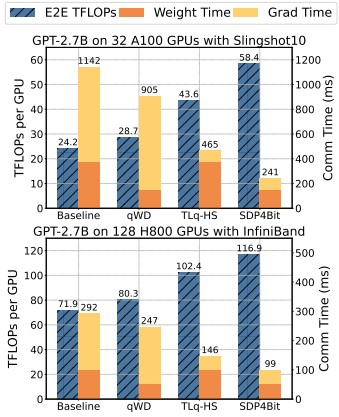

Figure 8: Throughput breakdown of SDP4Bit on GPT-2.7B.

In addition, we demonstrate the scalability of SDP4Bit using GPT models of 6.7B and 13B parameters, with tests conducted on up to 128 GPUs, as shown in Figure 7. Under low bandwidth conditions, SDP4Bit achieves an average speedup of 3.40× for the 6.7B model and 2.49× for the 13B model. In high-bandwidth InfiniBand environments, the speedup averages 3.08× for the 6.7B model and 3.73× for the 13B model. The comparatively lower speedup for the 13B model under low bandwidth conditions can be attributed to the introduction of pipeline parallelism, which diminishes the proportion of communication handled by ShardedDP. Overall, SDP4Bit consistently maintains stable speedup performance across various GPU numbers and network environments.

### 5.4 Ablation Study

**Components Breakdown.** Figure 8 demonstrates the throughput improvement of qWD, TLq-HS, and their combination (SDP4Bit) on two different platforms. qWD alone provides a speedup ranging from $1.1\times$ to $1.2\times$, while TLq-HS alone results in an E2E speedup of $1.4\times$ to $1.8\times$. The notable benefit from gradient quantization stems from the high communication overhead associated with Float32 gradients in baseline training, which is higher compared to BFloat16 weights. When they are applied together, SDP4Bit achieves a more substantial speedup, ranging from $1.6\times$ to $2.4\times$.

**TLq-HS vs. ULq.** Table 4 compares gradient quantization between TLq-HS and ULq. The results show that although TLq-HS employs 8-bit quantization for intra-node gradient communication, it introduces negligible overhead compared to 4-bit communication. This is due to 1) the high bandwidth of intra-node communication and 2) the fact that most intra-node communication is overlapped with the slower inter-node communication.

**Hadamard Kernel Fusion.** Table 4 shows that, compared to the SDP4Bit without fusing Hadamard Transform kernel, our optimized SDP4Bit reduces gradient communication overhead by 29%. Additionally, we provide a throughput comparison in Table 5 to further illustrate the impact of the Hadamard transformation. The results confirm that our Hadamard kernel fusion effectively reduces the overhead, making the transformation nearly zero-overhead and even matching the performance of quantization without the Hadamard transformation.

**Convergence with Different Group Sizes.** Table 3 examines the impact of various quantization granularities on the end-to-end validation loss during the pre-training of the GPT-125M model. For TLq-HS, a gradient quantization group size of 128 presents sufficient, with smaller sizes yielding no significant accuracy improvements. For qWD, a quantization group size of 2048 achieves training accuracy comparable to the baseline. Table 3 also presents the 4-bit weight quantization ($qW$) while using small group size. It is evident that even with very small group size (e.g., 32), direct 4-bit quantization leads to a significant gap in accuracy compared to the baseline, making 4-bit quantization for weights suboptimal.

## 6 Related Work

Apart from ZeRO++ [32] and QSDP [18], which are specifically designed for communication compression in ShardedDP, most previous studies have focused on traditional DP, primarily utilizing gradient compression. This includes both unbiased compression techniques [1, 33, 38, 5], which employ randomized compressors, and biased compression methods with error compensation [12, 31, 30, 29, 24] that require extra storage for residual errors, making them less suitable for resource-intensive training of LLMs. Other strategies like local optimization or federated learning reduce communication frequency rather than volume [16, 28, 35, 34, 2, 20], but increase memory use, complicating their application in LLM training. In addition, techniques like low-precision training [19, 22] and parameter-efficient fine-tuning [10, 3, 14] minimize the volume of trainable variables to reduce communication. In a different vein, weight quantization for inference has also been explored [7, 6, 39, 37, 4], employing more resource-intensive methods compared to those used in training to fine-tune compression parameters.

The Hadamard transform has been applied to machine learning data, as seen in HQ-MM's [36] compression of activations and THC's [15] gradient communication within a parameter server framework. Unlike THC, SDP4Bit enhances collective communication operations and GPU optimization.

## 7 Conclusion

In this paper, we propose SDP4Bit, a communication reduction strategy for Sharded Data Parallelism. SDP4Bit reduces both weight and gradient communication to nearly 4 bits while maintaining model accuracy comparable to the baseline. We implemented SDP4Bit in Megatron-LM and optimized it to reduce quantization overhead. Specifically, our experimental results demonstrate a training speedup of up to $4.08 \times$ on 128 GPUs. This paper focuses on LLM pre-training, but we plan to extend our work to other models and areas such as MoE, computer vision, and fine-tuning in the future.

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

# Appendix

## A Proofs

We use the following lemma (simplified from [27], Lemma 14) without proof.

**Lemma A.1.** *For every non-negative sequence $\{r_t\}_{t\geq 0}$ and any parameters $a \geq 0$, $c \geq 0$, $T \geq 0$, there exists a constant $\eta \leq \frac{1}{a}$, such that*

$$\frac{1}{T+1}\sum_{t=0}^{T}\left(\frac{r_t - r_{t+1}}{\eta} + c\eta\right) = \frac{1}{T+1}\frac{r_0 - r_{T+1}}{\eta} + c\eta \leq \frac{ar_0}{T+1} + \frac{2\sqrt{cr_0}}{\sqrt{T+1}}.$$

**Theorem 4.1** (Convergence error bound). *For arbitrary non-convex function under Assumption 4.1 and Assumption 4.2, taking learning rate $\eta \leq \frac{1}{10L\left(\frac{2}{\delta}+\rho\kappa+\rho+\kappa\right)}$, Algorithm 4 converges to a critical point with the following error bound:*

$$\frac{\sum_{t=0}^{T}\mathbb{E}[\|\nabla f(\tilde{w}_t)\|^2]}{T+1} \leq \frac{80L\left(\frac{2}{\delta}+\rho\kappa+\rho+\kappa\right)(f(w_0)-f^*)}{T+1} + 4\sigma\sqrt{\frac{(11-\delta)(\kappa+1)L(f(w_0)-f^*)}{T+1}}.$$

*Proof.* By using smoothness (Assumption 4.1), we have

$$f(w_{t+1}) \leq f(w_t) - \eta\langle\nabla f(w_t), \mathcal{U}_g(g_t)\rangle + \frac{\eta^2 L}{2}\|\mathcal{U}_g(g_t)\|^2.$$

Taking expectation w.r.t. the random compressor $\mathcal{U}_g$, we have

$$
\begin{aligned}
&\mathbb{E}_{gc}[f(w_{t+1})]\\
&\leq f(w_t) - \eta\langle\nabla f(w_t), g_t\rangle + \frac{\eta^2 L}{2}\mathbb{E}_{gc}\|\mathcal{U}_g(g_t)\|^2\\
&= f(w_t) - \eta\langle\nabla f(w_t), g_t\rangle + \frac{\eta^2 L}{2}[\|g_t\|^2 + \mathbb{E}_{gc}\|\mathcal{U}_g(g_t)-g_t\|^2]\\
&\leq f(w_t) - \eta\langle\nabla f(w_t), g_t\rangle + \frac{\eta^2 L(\kappa+1)}{2}\|g_t\|^2.
\end{aligned}
$$

Conditional on $w_t$, taking expectation on the random sample $\zeta_t$, we have

$$\mathbb{E}_\zeta[\mathbb{E}_{gc}[f(w_{t+1})]]$$

$$\leq f(w_t) - \eta\langle\nabla f(w_t), \nabla f(\tilde{w}_t)\rangle + \frac{\eta^2 L(\kappa+1)}{2}\mathbb{E}_\zeta\|g_t\|^2$$

$$= f(w_t) - \eta\langle\nabla f(w_t), \nabla f(\tilde{w}_t)\rangle + \frac{\eta^2 L(\kappa+1)}{2}\mathbb{E}_\zeta\|g_t - \nabla f(\tilde{w}_t) + \nabla f(\tilde{w}_t)\|^2$$

$$= f(w_t) - \eta\langle\nabla f(w_t), \nabla f(\tilde{w}_t)\rangle + \frac{\eta^2 L(\kappa+1)}{2}\mathbb{E}_\zeta[\|g_t - \nabla f(\tilde{w}_t)\|^2 + \|\nabla f(\tilde{w}_t)\|^2]$$

$$\leq f(w_t) - \eta\langle\nabla f(w_t), \nabla f(\tilde{w}_t)\rangle + \frac{\eta^2 L(\kappa+1)(\rho+1)}{2}\|\nabla f(\tilde{w}_t)\|^2 + \frac{\eta^2 L(\kappa+1)\sigma^2}{2}$$

$$\leq f(w_t) - \eta\langle\nabla f(w_t), \nabla f(\tilde{w}_t)\rangle + \frac{\eta^2 L(\kappa+1)(\rho+1)}{2}\|\nabla f(\tilde{w}_t)\|^2 + \frac{\eta^2 L(\kappa+1)\sigma^2}{2}$$

$$= f(w_t) - \eta\langle\nabla f(w_t) - \nabla f(\tilde{w}_t) + \nabla f(\tilde{w}_t), \nabla f(\tilde{w}_t)\rangle + \frac{\eta^2 L(\kappa+1)(\rho+1)}{2}\|\nabla f(\tilde{w}_t)\|^2$$

$$+ \frac{\eta^2 L(\kappa+1)\sigma^2}{2}$$

$$= f(w_t) - \eta\langle\nabla f(w_t) - \nabla f(\tilde{w}_t), \nabla f(\tilde{w}_t)\rangle - \eta\|\nabla f(\tilde{w}_t)\|^2 + \frac{\eta^2 L(\kappa+1)(\rho+1)}{2}\|\nabla f(\tilde{w}_t)\|^2$$

$$+ \frac{\eta^2 L(\kappa+1)\sigma^2}{2}$$

$$= f(w_t) - \eta\left(1 - \frac{\eta L(\kappa+1)(\rho+1)}{2}\right)\|\nabla f(\tilde{w}_t)\|^2 - \eta\langle\nabla f(w_t) - \nabla f(\tilde{w}_t), \nabla f(\tilde{w}_t)\rangle$$

$$+ \frac{\eta^2 L(\kappa+1)\sigma^2}{2}$$

$$\leq f(w_t) - \eta\left(1 - \frac{\eta L(\kappa+1)(\rho+1)}{2}\right)\|\nabla f(\tilde{w}_t)\|^2 + \frac{\eta}{2}\|\nabla f(w_t) - \nabla f(\tilde{w}_t)\|^2$$

$$+ \frac{\eta}{2}\|\nabla f(\tilde{w}_t)\|^2 + \frac{\eta^2 L(\kappa+1)\sigma^2}{2} \qquad\qquad \triangleright \langle a, b\rangle \leq \tfrac{1}{2}\|a\|^2 + \tfrac{1}{2}\|b\|^2$$

$$= f(w_t) - \frac{\eta}{2}\left[1 - \eta L(\kappa+1)(\rho+1)\right]\|\nabla f(\tilde{w}_t)\|^2 + \frac{\eta}{2}\|\nabla f(w_t) - \nabla f(\tilde{w}_t)\|^2 + \frac{\eta^2 L(\kappa+1)\sigma^2}{2}.$$

Again using smoothness, and taking $\eta \leq \frac{1}{2L(\rho+1)(\kappa+1)}$, we have $-\frac{\eta}{2}\left[1 - \eta L(\kappa+1)(\rho+1)\right] \leq -\frac{\eta}{4}$, and, we have

$$\mathbb{E}_\zeta[\mathbb{E}_{gc}[f(w_{t+1})]]$$

$$\leq f(w_t) - \frac{\eta}{2}\left[1 - \eta L(\kappa+1)(\rho+1)\right]\|\nabla f(\tilde{w}_t)\|^2 + \frac{\eta L^2}{2}\|w_t - \tilde{w}_t\|^2 + \frac{\eta^2 L(\kappa+1)\sigma^2}{2}$$

$$\leq f(w_t) - \frac{\eta}{2}\left[1 - \eta L(\kappa+1)(\rho+1)\right]\|\nabla f(\tilde{w}_t)\|^2 + \frac{\eta L^2}{2}\|e_t\|^2 + \frac{\eta^2 L(\kappa+1)\sigma^2}{2}$$

$$\leq f(w_t) - \frac{\eta}{4}\|\nabla f(\tilde{w}_t)\|^2 + \frac{\eta L^2}{2}\|e_t\|^2 + \frac{\eta^2 L(\kappa+1)\sigma^2}{2},$$

where we define the sequence

$$e_t = w_t - \tilde{w}_t, \quad e_0 = 0.$$

Now we establish the upper bound of the sequence $\|e_t\|^2$ as follows.

First, using $w_{t+1} = w_t - \eta\mathcal{U}_g(g_t)$ and $\tilde{w}_{t+1} = \tilde{w}_t + \mathcal{C}_w(w_{t+1} - \tilde{w}_t)$, we have the following equations:

$$w_{t+1} - \tilde{w}_{t+1} = e_{t+1} = w_t - \tilde{w}_t - \eta\mathcal{U}_g(g_t) - \mathcal{C}_w(w_{t+1} - \tilde{w}_t) = e_t - \eta\mathcal{U}_g(g_t) - \mathcal{C}_w(e_t - \eta\mathcal{U}_g(g_t))$$

Taking expectation w.r.t. the random compressor $\mathcal{C}_w$, we have

$$
\begin{aligned}
&\mathbb{E}_{wc}[\|e_{t+1}\|^2] \\
&= \mathbb{E}_{wc}[\|e_t - \eta\mathcal{U}_g(g_t) - \mathcal{C}_w(e_t - \eta\mathcal{U}_g(g_t))\|^2] \\
&\leq (1-\delta)\|e_t - \eta\mathcal{U}_g(g_t)\|^2.
\end{aligned}
$$

Taking expectation w.r.t. the random compressor $\mathcal{U}_g$, we have

$$
\begin{aligned}
&\mathbb{E}_{gc}[\mathbb{E}_{wc}[\|e_{t+1}\|^2]] \\
&\leq (1-\delta)\mathbb{E}_{gc}[\|e_t - \eta\mathcal{U}_g(g_t)\|^2] \\
&= (1-\delta)\mathbb{E}_{gc}[\|e_t - \eta g_t + \eta g_t - \eta\mathcal{U}_g(g_t)\|^2] \\
&= (1-\delta)\|e_t - \eta g_t\|^2 + (1-\delta)\eta^2\mathbb{E}_{gc}[\|g_t - \mathcal{U}_g(g_t)\|^2] \\
&\leq (1-\delta)\|e_t - \eta g_t\|^2 + (1-\delta)\eta^2\kappa\|g_t\|^2.
\end{aligned}
$$

Conditional on $w_t$, taking expectation on the random sample $\zeta_t$, we have

$$
\begin{aligned}
&\mathbb{E}_\zeta[\mathbb{E}_{gc}[\mathbb{E}_{wc}[\|e_{t+1}\|^2]]] \\
&\leq (1-\delta)\mathbb{E}_\zeta[\|e_t - \eta\nabla f(\tilde{w}_t) + \eta\nabla f(\tilde{w}_t) - \eta g_t\|^2] + (1-\delta)\eta^2\kappa\mathbb{E}_\zeta[\|g_t - \nabla f(\tilde{w}_t) + \nabla f(\tilde{w}_t)\|^2] \\
&= (1-\delta)\|e_t - \eta\nabla f(\tilde{w}_t)\|^2 + (1-\delta)(\kappa+1)\eta^2\mathbb{E}_\zeta[\|g_t - \nabla f(\tilde{w}_t)\|^2] + (1-\delta)\eta^2\kappa\|\nabla f(\tilde{w}_t)\|^2 \\
&\leq (1-\delta)\|e_t - \eta\nabla f(\tilde{w}_t)\|^2 + (1-\delta)(\kappa+1)\eta^2(\rho\|\nabla f(\tilde{w}_t)\|^2 + \sigma^2) + (1-\delta)\eta^2\kappa\|\nabla f(\tilde{w}_t)\|^2 \\
&= (1-\delta)\|e_t - \eta\nabla f(\tilde{w}_t)\|^2 + (1-\delta)\eta^2(\rho\kappa + \rho + \kappa)\|\nabla f(\tilde{w}_t)\|^2 + (1-\delta)(\kappa+1)\eta^2\sigma^2.
\end{aligned}
$$

With $\forall b > 0$, we have

$$
\begin{aligned}
&\mathbb{E}_\zeta[\mathbb{E}_{gc}[\mathbb{E}_{wc}[\|e_{t+1}\|^2]]] \\
&\leq (1-\delta)(1+b)\|e_t\|^2 + (1-\delta)(1+b^{-1})\|\eta\nabla f(\tilde{w}_t)\|^2 + (1-\delta)\eta^2(\rho\kappa + \rho + \kappa)\|\nabla f(\tilde{w}_t)\|^2 \\
&\quad + (1-\delta)(\kappa+1)\eta^2\sigma^2 \\
&= (1-\delta)(1+b)\|e_t\|^2 + (1-\delta)\eta^2[1 + b^{-1} + (\rho\kappa + \rho + \kappa)]\|\nabla f(\tilde{w}_t)\|^2 + (1-\delta)(\kappa+1)\eta^2\sigma^2.
\end{aligned}
$$

Then, by taking $b = \frac{\delta}{2(1-\delta)}$, we have $(1-\delta)(1+b) = 1 - \frac{\delta}{2}$, $1 + b^{-1} = \frac{2-\delta}{\delta} \leq \frac{2}{\delta}$, and

$$
\begin{aligned}
&\mathbb{E}_\zeta[\mathbb{E}_{gc}[\mathbb{E}_{wc}[\|e_{t+1}\|^2]]] \\
&\leq (1 - \frac{\delta}{2})\|e_t\|^2 + (1-\delta)\eta^2\left(\frac{2}{\delta} + \rho\kappa + \rho + \kappa\right)\|\nabla f(\tilde{w}_t)\|^2 + (1-\delta)(\kappa+1)\eta^2\sigma^2.
\end{aligned}
$$

We simplify the notation by denoting $\mathbb{E}[\|e_{t+1}\|^2] = \mathbb{E}_\zeta[\mathbb{E}_{gc}[\mathbb{E}_{wc}[\|e_{t+1}\|^2]]]$, and then unroll the sequence of $e_t$ back to $t = 0$.

$$
\begin{aligned}
&\mathbb{E}[\|e_{t+1}\|^2] \\
&\leq \sum_{\tau=0}^t (1 - \frac{\delta}{2})^{t-\tau}\left[(1-\delta)\eta^2\left(\frac{2}{\delta} + \rho\kappa + \rho + \kappa\right)\|\nabla f(\tilde{w}_\tau)\|^2 + (1-\delta)(\kappa+1)\eta^2\sigma^2\right] \\
&\leq (1-\delta)\eta^2\left(\frac{2}{\delta} + \rho\kappa + \rho + \kappa\right)\sum_{\tau=0}^t (1 - \frac{\delta}{2})^{t-\tau}\|\nabla f(\tilde{w}_\tau)\|^2 + (1-\delta)(\kappa+1)\eta^2\sigma^2\sum_{\tau=0}^t (1 - \frac{\delta}{2})^{t-\tau} \\
&\leq (1-\delta)\eta^2\left(\frac{2}{\delta} + \rho\kappa + \rho + \kappa\right)\sum_{\tau=0}^t (1 - \frac{\delta}{2})^{t-\tau}\|\nabla f(\tilde{w}_\tau)\|^2 + \frac{2(1-\delta)(\kappa+1)\eta^2\sigma^2}{\delta}.
\end{aligned}
$$

$$
\triangleright \sum_{\tau=0}^t (1 - \frac{\delta}{2})^{t-\tau} \leq \frac{1}{1-(1-\frac{\delta}{2})}
$$

Taking $\eta \leq \frac{1}{10L\left(\frac{2}{\delta} + \rho\kappa + \rho + \kappa\right)}$, we have

$$\mathbb{E}[\|e_{t+1}\|^2]$$

$$\leq \frac{1-\delta}{100L^2\left(\frac{2}{\delta} + \rho\kappa + \rho + \kappa\right)}\sum_{\tau=0}^{t}(1-\frac{\delta}{2})^{t-\tau}\|\nabla f(\tilde{w}_\tau)\|^2 + \frac{2(1-\delta)(\kappa+1)\eta\sigma^2}{\delta 10L\left(\frac{2}{\delta} + \rho\kappa + \rho + \kappa\right)}$$

$$\leq \frac{1-\delta}{100L^2\frac{2}{\delta}}\sum_{\tau=0}^{t}(1-\frac{\delta}{2})^{t-\tau}\|\nabla f(\tilde{w}_\tau)\|^2 + \frac{2(1-\delta)(\kappa+1)\eta\sigma^2}{\delta 10L\frac{2}{\delta}}$$

$$\leq \frac{(1-\delta)\delta}{200L^2}\sum_{\tau=0}^{t}(1-\frac{\delta}{2})^{t-\tau}\|\nabla f(\tilde{w}_\tau)\|^2 + \frac{(1-\delta)(\kappa+1)\eta\sigma^2}{10L}.$$

Then, stacking $\mathbb{E}[\|e_t\|^2]$ and taking total expectation, we have

$$\sum_{t=0}^{T}\mathbb{E}[\|e_{t+1}\|^2]$$

$$\leq \frac{(1-\delta)\delta}{200L^2}\sum_{t=0}^{T}\sum_{\tau=0}^{t}(1-\frac{\delta}{2})^{t-\tau}\|\nabla f(\tilde{w}_\tau)\|^2 + \frac{(T+1)(1-\delta)(\kappa+1)\eta\sigma^2}{10L}$$

$$\leq \frac{(1-\delta)\delta}{200L^2}\sum_{t=0}^{T}\left[\sum_{\tau=0}^{+\infty}(1-\frac{\delta}{2})^\tau\right]\|\nabla f(\tilde{w}_t)\|^2 + \frac{(T+1)(1-\delta)(\kappa+1)\eta\sigma^2}{10L}$$

$$\leq \frac{1-\delta}{100L^2}\sum_{t=0}^{T}\|\nabla f(\tilde{w}_t)\|^2 + \frac{(T+1)(1-\delta)(\kappa+1)\eta\sigma^2}{10L}.$$

Putting all the ingredients together and taking total expectation, we have

$$\sum_{t=0}^{T}\mathbb{E}[f(w_{t+1})]$$

$$\leq \sum_{t=0}^{T}\mathbb{E}[f(w_t)] - \frac{\eta}{4}\sum_{t=0}^{T}\mathbb{E}[\|\nabla f(\tilde{w}_t)\|^2] + \frac{\eta L^2}{2}\sum_{t=0}^{T}\mathbb{E}[\|e_t\|^2] + \frac{(T+1)\eta^2 L(\kappa+1)\sigma^2}{2}$$

$$\Rightarrow \quad \mathbb{E}[f(w_{T+1})]$$

$$\leq \mathbb{E}[f(w_0)] - \frac{\eta}{4}\sum_{t=0}^{T}\mathbb{E}[\|\nabla f(\tilde{w}_t)\|^2] + \frac{\eta L^2}{2}\sum_{t=0}^{T}\mathbb{E}[\|e_t\|^2] + \frac{(T+1)\eta^2 L(\kappa+1)\sigma^2}{2}$$

$$\Rightarrow \quad \mathbb{E}[f(w_{T+1})]$$

$$\leq \mathbb{E}[f(w_0)] - \frac{\eta}{4}\sum_{t=0}^{T}\mathbb{E}[\|\nabla f(\tilde{w}_t)\|^2] + \frac{\eta L^2}{2}\sum_{t=0}^{T}\mathbb{E}[\|e_t\|^2] + \frac{(T+1)\eta^2 L(\kappa+1)\sigma^2}{2}$$

$$\Rightarrow \quad \mathbb{E}[f(w_{T+1})]$$

$$\leq \mathbb{E}[f(w_0)] - \frac{\eta}{4}\sum_{t=0}^{T}\mathbb{E}[\|\nabla f(\tilde{w}_t)\|^2] + \frac{(1-\delta)\eta}{200}\sum_{t=0}^{T}\|\nabla f(\tilde{w}_t)\|^2$$

$$+ \frac{(T+1)(1-\delta)(\kappa+1)L\eta^2\sigma^2}{20} + \frac{(T+1)\eta^2 L(\kappa+1)\sigma^2}{2}$$

$$\Rightarrow \quad \mathbb{E}[f(w_{T+1})] \leq \mathbb{E}[f(w_0)] - \frac{\eta}{8}\sum_{t=0}^{T}\mathbb{E}[\|\nabla f(\tilde{w}_t)\|^2] + \frac{(T+1)(11-\delta)(\kappa+1)L\eta^2\sigma^2}{20}$$

$$\Rightarrow \quad \frac{1}{8(T+1)}\sum_{t=0}^{T}\mathbb{E}[\|\nabla f(\tilde{w}_t)\|^2] \leq \frac{1}{T+1}\frac{\mathbb{E}[f(w_0)] - \mathbb{E}[f(w_{T+1})]}{\eta} + \frac{(11-\delta)(\kappa+1)L\eta\sigma^2}{20}$$

Finally, using Lemma A.1, we have

$$\frac{1}{T+1}\sum_{t=0}^{T}\mathbb{E}[\|\nabla f(\tilde{w}_t)\|^2]$$

$$\leq \frac{8}{T+1}\frac{\mathbb{E}[f(w_0)]-f^*+f^*-\mathbb{E}[f(w_{T+1})]}{\eta}+\frac{8(11-\delta)(\kappa+1)L\eta\sigma^2}{20}$$

$$\leq \frac{80L\left(\frac{2}{\delta}+\rho\kappa+\rho+\kappa\right)(f(w_0)-f^*)}{T+1}+4\sigma\sqrt{\frac{(11-\delta)(\kappa+1)L(f(w_0)-f^*)}{T+1}}.$$

$\square$

## B   Other Evaluation Results

To further demonstrate the effectiveness of SDP4Bit in enhancing training efficiency, we present the relationship between wall clock time and training loss in Figure 9.

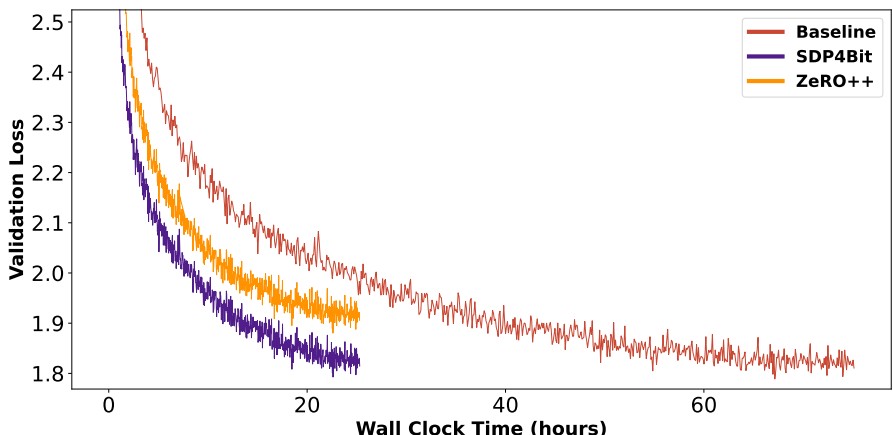

Figure 9: Comparison of validation loss versus wall-clock time for Baseline, ZeRO++ and SDP4Bit on the GPT-6.7B model.

To further illustrate the impact of the Hadamard transformation on (de)quantization performance, we provide (de)quantization throughput experiment in Table 5, which is tested on an A100 GPU.

| Input/Output Size | Quantization | | Dequantization | |
|---|---|---|---|---|
| | w/o Had. | w/ Had. | w/o Had. | w/ Had. |
| 8 MB | 305.6±10.9 | 301.8±10.6 | 367.7±10.6 | 359.6±9.6 |
| 16 MB | 389.0±12.8 | 387.1± 8.2 | 428.0±10.6 | 428.6±7.6 |
| 64 MB | 494.8± 3.7 | 493.7± 2.6 | 505.3± 2.1 | 505.6±2.2 |
| 512 MB | 682.1± 0.8 | 681.6± 1.2 | 685.1± 0.8 | 685.2±0.6 |
| 1024 MB | 686.5± 1.2 | 686.3± 0.4 | 688.0± 0.3 | 688.0±0.3 |
| 2048 MB | 688.6± 0.2 | 688.6± 0.2 | 689.5± 0.2 | 689.4±0.2 |

Table 5: (De)quantization Throughput with/without Hadamard, including std. dev.

## C   Notations in Training

| | |
|---|---|
| qW | original ZeRO++ int4 *weight* quantization |
| qWD | weight difference int4 quantization |
| ULq | original ZeRO++ uniform-level Int4-Int4 all-to-all *gradient* quantization |
| TLq | two-level Int8-Int4 all-to-all *gradient* quantization |
| TLq-HS | two-level Int8-Int4 all-to-all *gradient* quantization with Hadamard Smoother |

Table 6: Notations in experiments.

# D Detailed Training Settings

In the experimental section, we utilize a total of six different sizes of GPT models. Their model configurations are detailed in Table 7.

For the accuracy experiments, we standardize the batch size to 256, and set sequence length to 2048. We use AdamW [17] optimizer in all the experiments. The detailed training parameters are listed in Table 9.

In the throughput experiments, to more clearly study the communication bottleneck and ensure consistency across different GPU counts, we set the accumulation step to 1. The batch size is adjusted according to the number of GPUs, and the sequence length (micro batch) is uniformly set to 2048. Due to the different number of GPUs per node in the two architectures, we adjusted the tensor parallel size (TP) and pipeline parallel size (PP) accordingly, referencing [26], to achieve the highest throughput. Specifically, the maximum tensor parallel size is 4 for the 4xA100 environment and 8 for the 8xH800 environment. See detailed parameters in Table 8.

Table 7: Model Size Parameters

| Model Size | Sequence Length | Hidden Size | Layers |
|---|---|---|---|
| 125M | 2048 | 768 | 12 |
| 350M | 2048 | 1024 | 24 |
| 1.3B | 2048 | 2048 | 24 |
| 6.7B | 2048 | 4096 | 32 |
| 13B | 2048 | 5120 | 40 |
| 18B | 2048 | 6144 | 40 |

Table 8: Parallel Configuration for Throughput Test

| Model Size | TP | PP | Accumulation Step |
|---|---|---|---|
| 1.3B | 1 | 1 | 1 |
| 2.7B | 1 | 1 | 1 |
| 6.7B | 4 | 1 | 1 |
| 13B | 4/8 | 2/1 | 1 |
| 18B | 4/8 | 2/1 | 1 |

Table 9: E2E Convergence Training Parameters

| Model Size | Learning Rate | Betas | Epsilon | Weight Decay | Batch Size |
|---|---|---|---|---|---|
| 125M | 6e-4 | 0.9, 0.95 | 1e-8 | 0.1 | 256 |
| 350M | 3e-4 | 0.9, 0.95 | 1e-8 | 0.1 | 256 |
| 1.3B | 2e-4 | 0.9, 0.95 | 1e-8 | 0.1 | 256 |
| 6.7B | 12e-5 | 0.9, 0.95 | 1e-8 | 0.1 | 256 |

