# OpenReview forum: "SDP4Bit: Toward 4-bit Communication Quantization in Sharded Data Parallelism for LLM Training"
_NeurIPS.cc/2024/Conference — NeurIPS 2024 poster_

### Official Review · Reviewer_4e2h · 2024-07-07

**Soundness:** 3
**Presentation:** 3
**Contribution:** 3
**Rating:** 7
**Confidence:** 4

**Summary:**

This paper presents a novel scheme for reducing the communication cost of training LLMs using FSDP approach. To this end, the authors suggest to quantize the weights differences (instead of the weights themselves) and applying Hadamard transformations for making the gradients smooth. They show up to ~4x training speedup on 128 GPUs.

**Strengths:**

1. The paper studies a highly important problem (communication cost of llm training).
2. Although the ideas of quantizing the weights difference and Hadamard smoothing are not new, using them in this direction is interesting.
3. The scheme is evaluated on a solid setting (16x4 A100 and 16x8 H100).
4. The theoretical convergence analysis of difference quantization is nice.
5. The paper is well-written and easy to follow.

**Weaknesses:**

1. Although, the idea is interesting, I think the authors should provide more ablations on various aspects of their design choice (see questions section) to elaborate the approach better.
2. To motivate the problem more, I would like to see some practical analysis of the communication volume based on the number of nodes. For example, how much data (in GB) will be sent between gpus in a single node (and between different nodes) when we scale to a high number of gpus.

**Questions:**

I have some questions/ideas about the paper. I would be happy if the authors answer them.

1. The paper applies symmetric quantization for all schemes. However, in the communication reduction techniques, it is natural to use asymmetric quantization (which is a bit more accurate). Also, stochastic rounding could be another option to have unbiased gradients. Have you tried those schemes to see the accuracy changes? Maybe you don't need to apply hadamard in that case and the accuracy will be recovered!

2. The main motivation behind the hadamard smoothing is to remove the outliers. This is studied in several works [1, 2, 3, 4] (which are not cited here also). My main question is why we do not see an analysis of the gradient distribution in the paper so we can make sure that the hadamard is important.

3. There isn't any study on the overhead of applying Hadamard scheme. I agree that there are fast hadamard kernels when the input is power-of-two. However, it is not clear for me if your input is not power-of-two. Also, there isn't an ablation on the size of the Hadamard in that case.

References:

1. Training Transformers with 4-bit Integers (https://arxiv.org/abs/2306.11987)
2. QuaRot: Outlier-Free 4-Bit Inference in Rotated LLMs (https://arxiv.org/abs/2404.00456)
3. QuIP#: Even Better LLM Quantization with Hadamard Incoherence and Lattice Codebooks (https://arxiv.org/pdf/2402.04396)
4. SpinQuant: LLM quantization with learned rotations (https://arxiv.org/abs/2405.16406)

**Limitations:**

Yes. The author discuss that the work focuses only on communication reduction.

---

> ### Author Rebuttal · Authors · 2024-08-04
>
> ### Weaknesses
>
> **W2: To motivate the problem more, I would like to see some practical analysis of the communication volume based on the number of nodes. For example, how much data ...**
>
> A: The practical communication volume for data parallelism is primarily determined by factors such as model size, the number of GPUs, and the quantization strategy employed. We present the practical volume result for different models below. Note that for larger models (6.7B and above), tensor parallelism is employed, so the intra-node data-parallel size may not equal the number of GPUs within a node.
>
> #### Notation
> | Model Size | GPUs per Node | Nodes | Gradient Quant Group | Weight Quant Group |
> |--|----|--|--|--|
> | M          | g         | N     | G                 | W                    |
>
> #### Number of Bytes sent by a single GPU (when data parallel size = number of total GPUs, g > 1, N > 1)
>
> |  | Baseline &nbsp;&nbsp;&nbsp;&nbsp; &nbsp;&nbsp;&nbsp;&nbsp; | SDP4Bit |
> |-----|----|----|
> | **Weight**     |                         |                             |
> | Intra:         | $$2M\frac{gN - 1}{gN}$$ | $$(0.5M + 4\lceil M/W\rceil)\frac{gN - 1}{gN}$$ |
> | Inter:         | $$2M\frac{gN - 1}{gN}$$ | $$(0.5M + 4\lceil M/W\rceil)\frac{gN - 1}{gN}$$ |
> | **Gradient**   |                         |                             |
> | Intra:         | $$4M\frac{gN - 1}{gN}$$ | $$(M + 4\lceil M/G\rceil)\frac{g - 1}{g}$$       |
> | Inter:         | $$4M\frac{gN - 1}{gN}$$ | $$(0.5\frac{M}{g} + 4\lceil\frac{M}{gG}\rceil)\frac{N - 1}{N}g$$ |
>
> #### Number of Bytes sent per GPU per Iteration (32 A100 GPUs)
>
> | Model Size | Baseline Weight         | Baseline Gradient             | SDP4Bit Weight             | SDP4Bit Gradient           |
> |----|------|------|-------|------|
> | GPT-1.3B   | Intra: 2.37 GB, Inter: 2.37 GB | Intra: 4.75 GB, Inter: 4.75 GB | Intra: 0.60 GB, Inter: 0.60 GB | Intra: 0.95 GB, Inter: 1.15 GB |
> | GPT-2.7B   | Intra: 4.79 GB, Inter: 4.79 GB | Intra: 9.57 GB, Inter: 9.57 GB | Intra: 1.20 GB, Inter: 1.20 GB | Intra: 1.91 GB, Inter: 1.15 GB |
> | GPT-6.7B   | Intra: 0 GB, Inter: 2.72 GB | Intra: 0 GB, Inter: 5.45 GB | Intra: 0 GB, Inter: 0.68 GB | Intra: 0 GB, Inter: 0.72 GB |
> | GPT-13B    | Intra: 0 GB, Inter: 2.30 GB | Intra: 0 GB, Inter: 4.61 GB | Intra: 0 GB, Inter: 0.58 GB | Intra: 0 GB, Inter: 0.61 GB |
> | GPT-18B    | Intra: 0 GB, Inter: 3.29 GB | Intra: 0 GB, Inter: 6.58 GB | Intra: 0 GB, Inter: 0.83 GB | Intra: 0 GB, Inter: 0.87 GB |
>
>
> This data illustrates the communication volume involved in different model sizes and quantization strategies. Notably, larger models like GPT-6.7B and above also enable tensor parallelism which will decrease intra-node dp size, eliminating intra-node dp communication.
>
> ### Questions:
> **Q1**
>
> **A:** We appreciate the insightful question regarding the use of asymmetric quantization and stochastic rounding. In our experiments, we've applied symmetric quantization with stochastic rounding across all methods to ensure a fair comparison. Our choice aligns with the baseline papers such as ZeRO++ for consistent comparison.
>
> While we acknowledge that different compressors can have varying impacts on performance, our main contributions lie in the areas of weight difference quantization and two-level mixed-precision quantization. The exploration of different compressor types, such as asymmetric quantization or even sparsification and their potential benefits, is orthogonal to our work. Although investigating different compressors is valuable, it's beyond the scope of our current study. Also note that one of our main contributions in gradient quantization is the system optimization (kernel fusion, etc.) which could also be applied to asymmetric quantization.
>
> **Q2**
>
> **A:** We appreciate the suggestion to analyze gradient distribution to justify the use of the Hadamard Transform. We've added and additional analysis illustrated in **Figure 3 of our rebuttal PDF** for the gradient distribution. This analysis demonstrates the necessity of the Hadamard Transform in handling gradient outliers, thereby justifying its use in our methodology. We will also include the additional relevant works ([2, 3, 4]) to our revised manuscript, as well as the analysis of gradient distribution.
>
> **Q3**
>
> **A:** The overhead reduction is two-fold: the choice of a small Hadamard matrix size and kernel fusion. The experiments in Table 4 (TLq-HS vs. Int4-Int4) demonstrate that there is nearly no additional overhead due to these optimizations.
>
> As described in Section 3.3, our proposed TLq-HS uses a Hadamard matrix size of 32x32, which we consider the best trade-off between accuracy and performance. Although larger Hadamard matrices can better smooth outliers, the improvement is modest (**illustrated in the table below with different Hadamard size**). Additionally, larger matrices increase computational complexity, which can become a bottleneck, especially on older GPU versions, and can result in inefficient memory access patterns due to larger tiling sizes.
>
> Given the small size of the Hadamard matrix (32x32), the Hadamard transformation is primarily a memory-bound operation on contemporary GPUs. This characteristic allows us to seamlessly integrate the Hadamard transform with the quantization kernel, resulting in nearly zero additional overhead. To further illustrate the impact of the Hadamard transformation on performance, we provide (de)quantization throughput experiment in our **rebuttal pdf table 2**, which is tested on an A100 GPU.
>
> If the input size is not divisible by 32, we pad the input with zero accordingly.
>
> As requested, the ablation study of **validation loss with different Hadamard sizes** is provided below. We will also include these results in the revised version.
> |  GPT-125M   | Val Loss  |
> |-|-|
> | Baseline | 2.29392 |
> | TLq-HS (Hadamard Size=32)  | 2.29528|
> | TLq-HS (Hadamard Size=64)  | 2.29597|
> | TLq-HS (Hadamard Size=128) | 2.29612 |

---

> > ### Comment · Reviewer_4e2h · 2024-08-11
> > **Reply**
> >
> > Thanks for your helpful answers and explanation. I will raise my score to 7.

---

> > > ### Author Response · Authors · 2024-08-11
> > >
> > > Thank you for your insightful review and valuable advice, which have been very helpful in improving our paper.

---

### Official Review · Reviewer_6CdP · 2024-07-08

**Soundness:** 3
**Presentation:** 4
**Contribution:** 3
**Rating:** 5
**Confidence:** 4

**Summary:**

Thank you for submitting your paper to Neurips 2024. The paper proposes SDP4Bit, a communication quantization method that mitigates accuracy loss by weight difference quantization and 8-bit intra-node and 4-bit internode quantization. The authors provide convergence analysis to show that SDP4Bit is compatible with both biased and unbiased compressors and experimental results to show that SDP4Bit outperforms existing solutions.

**Strengths:**

1. The authors provide convergence analysis to show that weight difference quantization is compatible with both biased and unbiased compressors.
2. The two-level quantization method is suitable for the existing GPU cluster network topology.
3. The authors' implementation uses several system optimizations.

**Weaknesses:**

1. Missing key ablation studies. The two proposed strategies in this paper are weight difference quantization and two-level int8-int4 all-to-all gradient averaging. So I suggest the authors add the following two experiments to compare the accuracies.
- Running ZeRO++ with two-level int8-int4 all-to-all gradient averaging and its original quantization method
- Running ZeRO++ with its original int4-int4 all-to-all gradient averaging and the weight difference quantization.
2. The figures are too small to read.
3. Related work could add AG-SGD, which compresses changes in activations rather than the activations themselves, so it does not rely on the assumption that gradients are unbiased.
[1] Fine-tuning Language Models over Slow Networks using Activation Quantization with Guarantees

**Questions:**

1. Under what assumptions are weight differences generally easier to quantify? I've read several papers based on this intuition, but none that provide a comprehensive analysis of this intuition.
2. If you only need to assume that the magnitude of the weight difference is smaller than the weight, how do you ensure this?
3. If you ensure this via the learning rate, I think it will potentially lead to higher wall clock convergence time.

**Limitations:**

Yes.

---

> ### Author Rebuttal · Authors · 2024-08-04
>
> ### Weaknesses
>
> **W1: Missing key ablation studies. The two proposed strategies in this paper are weight difference quantization and two-level int8-int4 all-to-all gradient averaging. So I suggest the authors add the following two experiments to compare the accuracies.**
> - *Running ZeRO++ with two-level int8-int4 all-to-all gradient averaging and its original quantization method.*
> - *Running ZeRO++ with its original int4-int4 all-to-all gradient averaging and the weight difference quantization.*
>
>
> **A:**
> We do have ablation studies for weight-only quantization and gradient-only quantization in Table 1 and Figure 5 (and we apologize for any potential confusion caused by the abbreviations). By separating the evaluation of gradient and weight quantization strategies, we avoid the influence of quantization on other parts of communication, and provide a clear understanding of the effectiveness of our proposed method.
>
> Instead of the two suggested experiments, we have the following ones:
> - *Running with full-precision weight communication: compare the proposed two-level int8-int4-Hadamard all-to-all gradient averaging (denoted as TLq-HS) with the original ZeRO++ two-level int4-int4 all-to-all gradient averaging (denoted as Int4-Int4)* (**Figure 5 after Line 313**)
> - *Running with full-precision gradient communication: compare the proposed int4 weight difference quantization (denoted as qWD) with the original ZeRO++ int4 weight quantization (denoated as quant-W4)* (**Table 1 at the top of Page 8**)
>
> We also acknowledge that there was no direct comparison between our proposed gradient quantization and the original ZeRO++ gradient quantization without the influence of Hadamard transform. To address this, we include the Int8-Int4 all-to-all gradient averaging (denoted as TLq) results in **Figure 2 of rebuttal PDF**, which will also be included in the revised manuscript.
>
> **W2: The figures are too small to read.**
>
> **A:**
> We acknowledge the issue with the figures being too small to read. We will increase the size of the figures in the revised version to enhance readability.
>
> **W3: Related work could add AG-SGD, which compresses changes in activations rather than the activations themselves, so it does not rely on the assumption that gradients are unbiased. [1] Fine-tuning Language Models over Slow Networks using Activation Quantization with Guarantees**
>
> **A:**
> Thank you for pointing out the relevant work on AG-SGD, which compresses changes in activations rather than the activations themselves. We will include this work in our related work section.
>
> ### Questions
>
> **Q1: Under what assumptions are weight differences generally easier to quantize? I've read several papers based on this intuition, but none that provide a comprehensive analysis of this intuition.**
>
> **A:**
> In our paper, "easier to quantize" refers to achieving lower relative reconstruction error during quantization. This intuition is based on the observation on the real data, as illustrated in Figure 4 Page 4. The data distribution in Figure 4 shows that: 1) the quantization on weight differences achieves a finer granularity (smaller gaps between quantization levels) compared to quantization on weights themselves; 2) the weight differences are smaller than weights in magnitude or range, thus combining with the informal analysis in Line 136-143 of Page 4, we intuitively argue that weight differences potentially has lower reconstruction error relative to the weights themselves, which motivates our algorithm design.
>
> **Q2: If you only need to assume that the magnitude of the weight difference is smaller than the weight, how do you ensure this?**
>
> **Q3: If you ensure this via the learning rate, I think it will potentially lead to higher wall clock convergence time.**
>
> **A (for Q2 and Q3 together):**
> We do not assume nor ensure "the magnitude of the weight difference is smaller than the weight" at all, either in practice or in theoretical analysis. In fact, we use exactly the same learning rate as the non-compression baselines in the experiments.
>
> Note that the statements such as "magnitudes of weight differences are smaller than those of weights themselves" in Section 3.1 are simply an intuition and observation on real data that motivates weight difference quantization. It is not an assumption required in the actual theoretical analysis.
>
> We further explain such an intuition here: For any optimizer, it could be summarized as pesudo code: $w \leftarrow w - lr * update$, where $w$ is the weight parameter and $lr * update$ is exactly the weight difference. We typically expect that $lr * update$ is much smaller than $w$ itself, otherwise, the optimizer update would override the weight parameter entirely, which doesn't make sense.
>
> Also, note that we don't really need the assumption of relatively smaller weight differences to prove that weight difference compression is better than weight compression in theory. In fact, the gap between them is whether to converge or not. In Counterexample 4.1 Line 217, we show that under standard settings (smooth function and unbiased gradients) weight compression + SGD may not even converge, no matter how small the learning rate is. In contrast, we provide rigorous proof in Theorem 4.1 (Appendix A) showing that our proposed weight difference compression + SGD converges in the same rate as ordinary SGD.

---

> > ### Comment · Reviewer_6CdP · 2024-08-13
> >
> > Thank you for adding ablation studies and the explanation of "easier to quantize". I will maintain my score.

---

### Official Review · Reviewer_MiBn · 2024-07-19

**Soundness:** 3
**Presentation:** 3
**Contribution:** 3
**Rating:** 6
**Confidence:** 4

**Summary:**

This paper proposed a 4bit quantization framework for sharded data parallelism training. It proposed to quantize the weight differences between iterations as the first method to reduce the accuracy degradation. It also proposed to mixed-precision quantization for intra- and inter- node communication. Experiment results show that the proposed system can achieve 2x to 4x speedup on training GPT models with size from millions to a few billions.

**Strengths:**

1. The problem of communication bottleneck in FSDP, especially multi-node training is important. The paper aims to solve this problem and  the results seems to encourage this approach.

2. The paper implemented the framework in real hardware environment and achieves 2-4x speedup by reducing the communication.

**Weaknesses:**

1. The experiment results shows training/validation loss on Pile for the proposed method and baseline (FP) and QSDP and other method. The loss is an auxiliary variable to the performance of the model. A direct comparison on real dataset will be more convincing. For example, what is the accuracy of the model training by proposed methods and ZeRO/QSDP on mmlu and hellaswag/piqa/winogrande/ARC/ARE etc reasoning tasks, and gsk8k etc math tasks? This is critical to show the accuracy/precision improvement over other 4-bit communication methods.

2. The paper shows the speedup of the proposed methods, but I  would like to know which results in Table 4 is the training throughput for ZeRO++ and QSDP (if applicable)? I would like to see a comparison for accuracy-training_speed between the proposed method and ZeRO /QSDP. Furthermore, is there any results on the training throughput (tokens/sec) at certain sequence length (instead of training TFLOPs)?

3. The paper introduces a few method, including qWD, two-level quantization, Hadamard transforms, etc. I would like to know which ones are the invention of the paper? For example, using Hadamard transform to handle the outlier has been known for a while.

**Questions:**

Please refer to the session "Weakness"

---

> ### Author Rebuttal · Authors · 2024-08-04
>
> ### Weaknesses
>
> **W1: The experiment results show training/validation loss on Pile for the proposed method and baseline (FP), QSDP, and other methods. The loss is an auxiliary variable to the performance of the model. A direct comparison on real datasets would be more convincing. For example, what is the accuracy of the model trained by the proposed methods and ZeRO/QSDP on MMLU and Hellaswag/PIQA/Winogrande/ARC/ARE, etc., reasoning tasks, and GSK8K, etc., math tasks? This is critical to show the accuracy/precision improvement over other 4-bit communication methods.**
>
> **A:** We thank the reviewer for suggesting these evaluation methods for a more comprehensive comparison. We use validation loss as a measurement to align with our baseline papers ZeRO++/QSDP. Furthermore, we believe that the loss, as first-hand information, is one of the best metrics to reflect the influence of communication compression on the training procedure.
>
> **W2: The paper shows the speedup of the proposed methods, but I would like to know which results in Table 4 represent the training throughput for ZeRO++ and QSDP (if applicable)? ...**
>
> **A:**
> - **ZeRO++ Results in Table 4:** We thank the reviewer for highlighting this point. We acknowledge that the use of numerous abbreviations might have caused some confusion. The term "Int4-Int4" in Table 4, as defined on lines 277 and 292, represents the configuration used in ZeRO++ when gradient quantization is enabled. Specifically, "Int4-Int4" denotes 4-bit gradient compression for both intra-node and inter-node communication, which aligns with ZeRO++'s quantization strategy.
>
> - We did not include the throughput results for ZeRO++ with both weight and gradient quantization enabled because the resulting accuracy degradation was substantial. As shown in Figure 1 and line 291-292, the configuration with W4 (4-bit weight quantization) combined with Int4-Int4 (4-bit 2-level gradient quantization) exhibits a significant accuracy gap compared to the baseline, making it impractical for pre-training tasks where accuracy is critical.
>
> - **Accuracy-Training Speed Comparison:** The comparison of accuracy versus wall-clock time between our proposed SDP4Bit method and ZeRO++ (W4&Int4-Int4) for GPT-6.7B training on 128 H800 GPUs is presented in **Figure 1** of the rebuttal document.
>
> - **Training Throughput in Tokens/Sec:** While we primarily report training throughput in TFLOP/s, these results can also be presented in tokens/sec. The detailed corresponding results are provided below:
>
> | Model Size | 4xA100 Baseline (tokens/sec) | 4xA100 SDP4Bit (tokens/sec) | 4xA100 Speedup | 8xH800 Baseline (tokens/sec) | 8xH800 SDP4Bit (tokens/sec) | 8xH800 Speedup |
> |------------|---------|----------|---|---------|-----------|-----|
> | 1.3B       | 169,723.8                | 406,228.9               | 2.39$\times$           | 974,432.5                 | 1,498,441.4              | 1.54$\times$           |
> | 2.7B       | 85,886.9                  | 209,101.9               | 2.43$\times$           | 514,489.8                 | 836,600.1                | 1.63$\times$           |
> | 6.7B       | 63,950.4                  | 220,223.3                | 3.44$\times$           | 310,965.6                 | 923,946.0                  | 2.97$\times$           |
> | 13B        | 60,795.3                  | 161,877.3               | 2.66$\times$           | 172,901.7                 | 666,543.1                | 3.86$\times$           |
> | 18B        | 44,843.3                  | 130,724.9               | 2.92$\times$           | 126,892.3                 | 519,622.9                | 4.09$\times$           |
>
> **W3: The paper introduces a few methods, including qWD, two-level quantization, Hadamard transforms, etc. I would like to know which ones are the invention of the paper? ...**
>
> **A:**
> Our proposed SDP4Bit is a system-algorithm co-design where the qWD (quantization on Weight Difference) mainly contributes to the algorithm part, while the TLq-HS (Two-level gradient quantization and Hadamard Smoother) contributes to the system part.
>
> **Our primary innovation** is the introduction of the **qWD method**, which involves quantizing weight differences instead of the weights themselves. To the best of our knowledge, this approach is novel and has not been proposed in prior works. We have provided theoretical proof of its convergence. The qWD method significantly reduces the weight communication overhead from the state-of-the-art 8-bit level (used in QSDP and ZeRO++) to a 4-bit level without compromising accuracy.
>
> **For TLq**, we introduced a mixed-precision approach (Int8-Int4) to achieve an optimal trade-off between accuracy and speed. While ZeRO++ utilizes Int4-Int4 quantization, our method combines 8-bit and 4-bit quantization to enhance accuracy.
>
> **Hadamard transform:** We acknowledge that the Hadamard Transform is a known technique for handling outliers and does not constitute a novel contribution in itself. However, our work makes significant contributions in minimizing its overhead. We achieved this by:
>   1. Eliminating the need for two consecutive Hadamard Transforms, as discussed on page 5, lines 183-186.
>   2. Integrating a Hadamard Transform with nearly-zero overhead within the quantization kernel. This optimization is detailed in Section 3.3, lines 190-210, and is evidenced by the similar overheads reported for the Int4-Int4 and TLq-HS configurations in Table 4. We also include detailed throughput results tesed on A100 GPU in our **rebuttal PDF table 2**.
>
> **Comprehensive compression strategy for shardedDP:** Combining all three methods, we established a unique system-algorithm co-design communication pattern for the sharded data parallelism (shardedDP). Our design is crucial for the efficient and scalable training of large-scale language models (LLMs) and is the first to reduce communication overhead of both parameter weight and gradient in shardedDP to nearly the 4-bit level without compromising accuracy.

---

### Official Review · Reviewer_Ah9Z · 2024-07-19

**Soundness:** 3
**Presentation:** 3
**Contribution:** 3
**Rating:** 6
**Confidence:** 4

**Summary:**

The paper introduces a novel approach to reduce communication overhead in Sharded Data Parallelism (ShardedDP) for training large language models (LLMs). It proposes two key techniques: quantization on weight differences and two-level gradient smooth quantization, which effectively compress weights and gradients to nearly 4 bits without compromising training accuracy. The method is implemented within the Megatron-LM framework and achieves up to 4.08× speedup in end-to-end throughput on 128 GPUs. The paper also provides theoretical guarantees of convergence and demonstrates negligible impact on training loss for models with up to 6.7 billion parameters.

**Strengths:**

1. Weight difference and two-level gradient quantization are more effective at retaining accuracy and significantly improving throughput than baseline approaches

2. The authors present a complete story with a new quantization algorithm, accompanying GPU kernel implementations, and detailed evaluation/ablation over a range of models and baselines which together clearly illustrate all the benefits of each component of their approach.

**Weaknesses:**

1. The connection between the convergence analysis and the quantization schemes is not entirely clear. Specifically, it is not clear if the quantizers correspond to the biased/unbiased compressors in Algorithm 4 and if so which of them is biased and which is unbiased. It appears from lines 5 and 7 of algorithm 4 that TLq is unbiased and qWD is biased but I couldn't find any mention/proof of that, so I am not sure if that is the case or if I am missing the point entirely.

2. Most of the empirical analysis is limited to the older generation of GPT models, and it is possible that may not extend as well to modern models like Mistral series, Phi series, Orca series, Llama series etc as well as the newer generation of GPT models

**Questions:**

1. Why is 4-bit quantization needed? Is 8-bit quantization not sufficient in practice, especially since we don't expect foundation models to be trained too frequently?

2. Why is inter node communication quantized to 4 bits while intra node communication is quantized to 8 bits and not the other way around?

3. I think the Hadamard transform matrix might need to be post multiplied by a diagonal matrix with randomly chosen +1/-1 values on its diagonal for it to be effective at removing outliers and flattening the data (https://arxiv.org/abs/1011.1595). Is that what has been done here? If not, why?

---

> ### Author Rebuttal · Authors · 2024-07-31
>
> ### Weaknesses
>
> **W1: The connection between the convergence analysis and the quantization schemes is not entirely clear. Specifically, it is not clear if the quantizers correspond to the biased/unbiased compressors in Algorithm 4 and if so which of them is biased and which is unbiased. It appears from lines 5 and 7 of Algorithm 4 that TLq is unbiased and qWD is biased but I couldn't find any mention/proof of that, so I am not sure if that is the case or if I am missing the point entirely.**
>
> **A:** The definition (notation) of the compressors $\mathcal{U}$ and $\mathcal{C}$ can be found in Definitions 4.1 and 4.2 on Page 6, right below Algorithm 4. The compressor $\mathcal{U}$ in Algorithm 4 Line 5 and Definition 4.1 is unbiased, assuming $\mathbb{E}[\mathcal{U}(v)] = v$. The compressor $\mathcal{C}$ in Algorithm 4 Line 7 and Definition 4.2 is (potentially) biased, as it does not assume unbiasedness. In the proof, the usage of Definition 4.1 can be found in Line 476 and Line 484 of the Appendix A, while the usage of Definition 4.2 is found in Line 483 of the Appendix A.
>
> **W2: Most of the empirical analysis is limited to the older generation of GPT models, and it is possible that may not extend as well to modern models like Mistral series, Phi series, Orca series, Llama series, etc., as well as the newer generation of GPT models.**
>
> **A:** The reason that we chose to use the open-source GPT-2 is that the main purpose of this paper is to accelerate the pretraining of LLMs, which is more time-consuming and thus benefits more from communication compression compared to fine-tuning tasks. The hyperparameters of a GPT-2 baseline pretrained on the Pile dataset are readily available in open-sourced repositories such as GPT-Neo. However, for models like Mistral and Llama, they are typically used for fine-tuning rather than pretraining from scratch on public datasets. The corresponding recipes (hyperparameters) and datasets for pretraining these models are not publicly available, making it challenging to establish a fair baseline.
>
> ### Questions
>
> **Q1: Why is 4-bit quantization needed? Is 8-bit quantization not sufficient in practice, especially since we don't expect foundation models to be trained too frequently?**
>
> **A:** The necessity for 4-bit quantization arises primarily from the need to reduce communication overhead, especially in scenarios with limited bandwidth. The choice of quantization level is influenced by the physical hardware and network infrastructure available:
>
> - **Weak Hardware:** For example, some NVIDIA GPUs, particularly the more affordable variants, lack high-speed NVLink and instead rely on PCIe for intra-node communication. This setup can significantly benefit from higher compression rates to mitigate slower communication channels.
> - **Heavy Communication Overhead:** In our communication overhead breakdown experiment shown in Figure 7, we observed that communication time (1142ms) constituted 76.7% of the total overhead compared to computation time (347ms) when training the GPT-2.7B model on 32 A100 GPUs equipped with 100Gbps slingshot10 inter-node bandwidth. Even with 8-bit quantization, the communication overhead remains substantial, as detailed in the communication overhead results **provided in the table below**. Hence, further compression to 4-bit levels can provide additional reductions in overhead.
>
> Additionally, while full retraining of foundation models may not be frequent, there are scenarios of continual training where incremental improvements or adaptations to new data are necessary. In these cases, efficient communication remains critical, making 4-bit quantization a practical solution.
>
> | Quantization Scheme (Weight/Gradient) | Computation | Communication | Communication Overhead |
> |---------------------------------------|-------------|---------------|------------------------|
> | Baseline (BF16/FP32)                  | 347 ms      | 1142 ms       | 76.7%                  |
> | Quantize to 8-bit (Int8/Int8)         | 347 ms      | 420 ms        | 54.8%                  |
> | SDP4Bit (Int4/Int4)                   | 347 ms      | 241 ms        | 41.0%                  |
>
> **Q2: Why is inter-node communication quantized to 4 bits while intra-node communication is quantized to 8 bits and not the other way around?**
>
> **A:** Intra-node communication typically uses high-speed communication media such as NVLink or at least PCIe, which is typically much faster than inter-node networking. Since inter-node is slower, it naturally becomes the major bottleneck and thus requires a higher compression ratio.
>
> **Q3: I think the Hadamard transform matrix might need to be post-multiplied by a diagonal matrix with randomly chosen +1/-1 values on its diagonal for it to be effective at removing outliers and flattening the data ([source](https://arxiv.org/abs/1011.1595)). Is that what has been done here? If not, why?**
>
> **A:** We didn't multiply the Hadamard matrix by a random +1/-1 diagonal matrix in this paper. The flattening power mainly comes from the original Hadamard matrix itself, which mixes the neighboring elements in the input matrix. The diagonal matrix only adds some randomness to the transformation, which may potentially enhance the flattening ability slightly. However, it also results in additional communication overhead because the receiver of the compressed message also needs the same +1/-1 diagonal matrix as the sender for dequantization. Furthermore, the additional random diagonal matrix incurs some difficulty to the optimization of the fused cuda kernel of Hadamard transform and quantization/dequantization. Thus, we chose to use the simplest form of Hadamard matrix without randomness.
>
> ---

---

> > ### Comment · Reviewer_Ah9Z · 2024-08-13
> > **Response to rebuttal**
> >
> > Thank you for addressing my concerns. As I had already recommended accepting the paper, I will keep my score.

---

### Author Rebuttal · Authors · 2024-08-04

We appreciate the reviewers' critical assessment of our work. Below, we provide the relevant figures and results to address the questions and concerns raised.

**Contents in Rebuttal PDF**

1. Table of notations that explains the abbrevations of algorithms such as W4, Int4-Int4, TLq, etc.
2. Comparison of Accuracy versus Wall-Clock Time Results
3. Additional ablation Study: Direct comparison between our proposed Int8-Int4 gradient quantization (denoted as TLq) and the original ZeRO++ gradient quantization (denoted as Int4-Int4)
4. Analysis of Gradient Distribution Before and After Hadamard Transformation
5. Comparison of (De)Quantization Kernel Speed with and without Hadamard Transformation

---

### Decision · Program_Chairs · 2024-09-25

**Decision:**

Accept (poster)

**Comment:**

All reviewers have given favorable ratings for this paper, ranging from 5 to 7.
Average Rating: 6.00 (Min: 5, Max: 7)
The following strengths of the paper have been highlighted. However, as there are few strong recommendations for this paper, an acceptance as a poster presentation is considered appropriate.

1: This paper proposes an innovative approach to reduce communication overhead in large language model (LLM) training with Sharded Data Parallelism (ShardedDP). The proposed method introduces quantization based on weight differences and two-level smooth quantization, compressing communication to nearly 4 bits without significantly compromising accuracy.

2: The proposed method demonstrates up to a 4.08x increase in end-to-end throughput when training GPT models with up to 6.7 billion parameters. Additionally, theoretical convergence guarantees are provided, and the impact on training loss is confirmed to be minimal.

3: The proposed method involves a system and algorithm co-design that includes GPU kernel implementation and runtime optimization, minimizing computation overhead. This aspect is highly valued as it contributes not only theoretical ideas but also practical implementation and optimization.

4: The method has been evaluated at a scale of up to 128 GPUs and is applicable to models of varying sizes. By improving the efficiency of communication compression, the approach proves useful in a wide range of training scenarios.